# HIV-Related Knowledge and Practices among Asian and African Migrants Living in Australia: Results from a Cross-Sectional Survey and Qualitative Study

**DOI:** 10.3390/ijerph20054347

**Published:** 2023-02-28

**Authors:** Daniel Vujcich, Alison Reid, Graham Brown, Jo Durham, Rebecca Guy, Lisa Hartley, Limin Mao, Amy B. Mullens, Meagan Roberts, Roanna Lobo

**Affiliations:** 1School of Population Health, Curtin University, Bentley, WA 6102, Australia; 2Centre for Social Impact, UNSW, Sydney, NSW 2052, Australia; 3Centre for Healthcare Transformation, Australian Centre for Health Services Innovation, School of Public Health and Social Work, Queensland University of Technology, Kelvin Grove, QLD 4059, Australia; 4Kirby Institute, UNSW, Sydney, NSW 2052, Australia; 5Centre for Human Rights Education, Curtin University, Bentley, WA 6102, Australia; 6Centre for Social Research in Health, UNSW, Sydney, NSW 2052, Australia; 7School of Psychology & Wellbeing, Centre for Health Research, Institute for Resilient Regions, University of Southern Qld, Ipswich, QLD 4305, Australia

**Keywords:** HIV, sexual health, Australia, Africa, Asia, migrants, ethnic groups, surveys and questionnaires, interviews, focus group discussions

## Abstract

Australian HIV notification rates are higher for people born in Northeast Asia, Southeast Asia and sub-Saharan Africa compared to Australian-born people. The Migrant Blood-Borne Virus and Sexual Health Survey represents the first attempt to build the national evidence base regarding HIV knowledge, risk behaviors and testing among migrants in Australia. To inform survey development, preliminary qualitative research was conducted with a convenience sample of n = 23 migrants. A survey was developed with reference to the qualitative data and existing survey instruments. Non-probability sampling of adults born in Northeast Asia, Southeast Asia and sub-Saharan Africa was undertaken (n = 1489), and descriptive and bivariate analyses of data were conducted. Knowledge of pre-exposure prophylaxis was low (15.59%), and condom use at last sexual encounter was reported by 56.63% of respondents engaging in casual sex, and 51.80% of respondents reported multiple sexual partners. Less than one-third (31.33%) of respondents reported testing for any sexually transmitted infection or blood-borne virus in the previous two years and, of these, less than half (45.95%) tested for HIV. Confusion surrounding HIV testing practices was reported. These findings identify policy interventions and service improvements critically needed to reduce widening disparities regarding HIV in Australia.

## 1. Introduction

While Australia’s public health response to human immunodeficiency virus (HIV) is widely celebrated as a ‘success story’ [1], there is growing recognition that this success has not been evenly distributed. Epidemiological data from 2019 show that Australian HIV notification rates were higher for people born in Northeast Asia (NEA; 3.5 per 100,000), Southeast Asia (SEA; 10.5 per 100,000) and sub-Saharan Africa (SSA; 8.5 per 100,000) compared to Australian-born people (2.6 per 100,000) [2]. Among people living with HIV in Australia, migrants have lower rates of diagnosis, antiretroviral therapy coverage, and viral suppression compared to Australian-born counterparts [3]. 

Given these salient disparities, culturally and linguistically diverse people from high-HIV-prevalence countries are recognized as a priority population in the Australian Government’s Eighth National HIV Strategy, *2018*–*2022* (hereafter, the Strategy) [4]. Key action areas under the Strategy are to: (1) “improve knowledge and awareness of HIV in priority populations and reduce risk behaviors associated with HIV transmission”; and (2) “improve the frequency, regularity and targeting of testing for priority populations” [4] (pp. 24–26). However, at the time the Strategy was developed, there was no systematic mechanism for measuring progress against these key action areas. 

Efforts to understand migrants’ HIV knowledge, attitudes and practices in Australia have largely been in the form of “short-term, small-scale projects and research studies” [5], none of which identified to date have been national in focus. For instance, a 2006–2008 survey collected data from 286 Thai, Cambodian, Sudanese, and Ethiopian migrants in Sydney, and a 2012 New South Wales survey collected data from 1406 migrants from Thailand, Cambodia and four African countries [6,7]. Other surveys of sexual health knowledge, attitudes and beliefs have been conducted with Sudanese youth in Queensland (n = 248) [8], gay Asian men in Australia (n = 604 in 2015/6 to 970 in 2021) [9], Western Australian residents born in SSA and SEA (n = 209) [10] and West African women who had recently migrated to Western Australia (n = 51) [11]. Direct comparison of the data obtained from these studies is not possible due to the use of different inclusion criteria, methods of recruitment and survey instruments. 

Thus, the aim of the Migrant Blood-Borne Virus and Sexual Health Survey (MiBSS) was to investigate the feasibility of developing and administering a periodic survey to migrants born in NEA, SEA and SSA living across Australia, with a view to building the national evidence base regarding HIV knowledge, risk behaviors and testing practices in a more comprehensive and consistent manner. As part of the survey development process, in-depth, semi-structured interviews and focus group discussions were conducted regarding access to sexual health and blood-borne virus services to identify concepts that merited further quantitative exploration. 

While preliminary qualitative research and the subsequent survey captured important information relating to a range of sexually transmissible infections (STIs) and blood-borne viruses (BBVs) including viral hepatitis, the focus of this article will be on presenting HIV-specific findings to answer the following questions: (1) What are NEA, SEA and SSA migrants’ knowledge, attitudes and practices in relation to HIV prevention and HIV testing in Australia; and (2) What are the barriers to and enablers of HIV prevention and HIV testing practices among NEA, SEA and SSA migrants living in Australia? Although the focus is on three regions of birth, it is recognized that each region contains a great deal of cultural and linguistic diversity. 

## 2. Materials and Methods

In light of the data regarding disparities between migrant and non-migrant Australians in HIV cascades of care, [3] preliminary qualitative research was conducted in 2019 to understand how migrants in Australia perceive and engage with services and resources relating to STIs and BBVs, including HIV. The purpose of this research was to ensure that relevant enablers of and barriers to health help-seeking were identified for inclusion in the subsequent MiBSS survey instrument. Adults born in NEA, SEA and SSA were recruited through convenience sampling in Western Australia and Queensland, with assistance from community and industry partner organizations with established links in migrant communities. Focus group discussions and one-on-one interviews were facilitated by AR, DV, MR, JD and AM. Participants were provided with an AUD 30 honorarium in recognition of their time (approximately 1.5 h). Informed consent was provided and discussions were audio-recorded. 

A Corbin–Straussian grounded theory approach was applied [12]. Grounded theory “is concerned with psycho-social processes of behavior and seeks to identify and explain how and why people behave in certain ways, in similar and different contexts” [13] (p. 1197). Corbin and Strauss’ approach recognizes that knowledge is not simply discovered—rather, knowledge is created by researchers working closely with the data in a manner that is transparent, systematic and self-reflexive [12,13]. Consequently, an initial topic guide was developed and used to guide data collection with three participants; the topic guide provided opportunities for participants to speak broadly about their experiences and perceptions of accessing STI and BBV services and resources both in Australia and in their countries of origin. Data were then reviewed by MR and DV and memoranda capturing initial reflections of emerging themes were drafted and used to inform revisions to the topic guide for use in subsequent interviews and focus group discussions (Appendix A). 

A professional service was used to transcribe the interview/focus group recordings, and the accuracy of the transcripts was checked against the audio recording by members of the research team. Two authors (MR and DV) reviewed transcripts and independently developed a coding matrix to reflect emerging themes and concepts; the matrices were compared in a meeting and a single matrix was agreed upon through consensus decision making. MR and DV then piloted the matrix using NVivo 12 to independently code the transcripts of four interviews and one focus group discussion [14]. Each coder drafted a memo to provide a transparent account of why data were coded in certain ways, and to document reasons for refining categories/sub-categories and re-categorizing data. The matrix was then refined by consensus and used by MR to code all remaining transcripts. 

An English-language survey instrument was then developed with reference both to the themes identified in the qualitative data and existing survey instruments [15]. The draft survey was pretested iteratively over three rounds with 62 NEA-, SEA- and SSA-born migrants living in four Australian states, as described by Vujcich and colleagues [15]. The survey was also translated into Khmer, Vietnamese, Karen, and Traditional Chinese using the TRAPD (Translation, Review, Adjudication, Pretesting and Documentation) model (see [16] for details of the process adopted, and rationale for language choices). A Simplified Chinese version was also offered. English-language and translated surveys were made available online and in paper form, in line with recommendations to use mixed-modes of data collection “to overcome the limitations associated with using each mode in isolation” [17] (p. 22). A full version of the print English survey can be accessed at https://www.mibss.org/publications (accessed on 10 November 2022). The online survey was created using Qualtrics software (Qualtrics, Provo, UT, USA) [18]. 

The survey sample comprised adults born in NEA, SEA and SSA and living in Australia. To explore inter-group variations, the aim was to recruit 372 respondents per birth region. Non-probability sampling techniques were adopted, which is consistent with other cross-sectional national HIV surveys in migrant communities such as the BASS Line study in England, and previous Australian studies [6,7,8,9,10,11,19]. The limitations of non-probability sampling are set out in the Discussion; however, the strength of the approach is that it creates opportunities to overcome poor response rates and inefficiencies associated with random, systematic or stratified sampling methods such as postal surveys or ‘cold calling’, particularly in the absence of adequate sampling frames or when seeking to access populations who may be wary of surveillance [20,21].

The approach taken to survey recruitment is described elsewhere in this special issue of *IJERPH* [22]. In summary, the recruitment process was primarily driven by organizations with experience working with migrant populations in Western Australia, South Australia, Victoria and Queensland. Recruitment strategies included: (1) face-to-face intercept at community events; (2) advertisements in diasporic newspapers and websites; (3) posters in locations frequented by migrants born in NEA, SEA and SSA; (4) Facebook posts, including boosted advertisements; and (5) personal invitations by phone call, text message or email [22]. Recruitment was primarily undertaken between September 2020 and May 2021 (the period reported in [22]), with an additional period of paper survey recruitment in Victoria extending to February 2022 due to that state’s unique experience of the COVID-19 pandemic and associated lockdowns.

Surveys were excluded from analysis if: (1) they were identified as having less than 25% progress in Qualtrics “on the assumption that these respondents were exploring the survey without intending to formally participate” [22]; (2) respondents indicated they “do not agree” to the statements in the consent form; (3) respondents reported they were not born in NEA, SEA or SSA, were not living in Australia, or were under the age of 18. 

Data were analyzed using Stata statistical software version 16.1 (StataCorp, College Station, TX, USA) [23]. Appendix A maps the variables created against survey questions and response options. All responses with an ‘other—open text’ option were reviewed to identify whether the response could be assigned to existing response options. Invalid responses (i.e., when skip logic was not followed or where more than one response was selected for a single-response question) were treated as skipped/missing data. 

Basic descriptive statistical analyses (frequency and percentage without standard errors or confidence intervals) were used to report demographic characteristics of the sample, and responses to knowledge, attitude, and practice questions. Chi-squared tests were used to compare knowledge, attitude, and practice question responses between groups by region of birth, gender, age, socio-economic status and time spent in Australia. 

The threshold for statistical significance in the reported findings is *p* ≤ 0.05. Qualitative findings are presented as illustrative quotations where they relate to, or assist in the interpretation of, survey findings relating to HIV prevention and HIV testing practices.

## 3. Results

### 3.1. Demographics

The survey sample comprised 1489 respondents, of whom 30.36% (n = 452) resided in Queensland, 28.01% (n = 417) resided in South Australia, 20.69% (n = 308) resided in Western Australia, 14.30% resided in Victoria (n = 213) and the remainder (6.65%; n = 99) were either from Australian states in which in-person recruitment was not conducted or did not provide sufficient data to enable state of residence to be determined. Respondents who identified as women represented the majority of the sample (59.10%; n = 880), with respondents who identified as men comprising 35.19% (n = 524). Twelve respondents (0.81%) either identified as both men and women, non-binary or ‘other’, and the remainder of the sample (4.90%; n = 73) did not provide valid responses to the gender survey item. Of the respondents who identified as men and answered the sexuality survey item (34.05%; n = 507), almost one in ten (9.86%; n = 50) reported being sexually attracted to other men (either exclusively or in addition to other genders). 

Over half of the survey sample reported being under the age of 40 (55.41%; n = 825), and the mean number of years spent living in Australia was 11.88 (SD 9.27, 114 respondents did not respond to this survey item). Almost three-quarters of respondents who answered the question regarding visa status reported being citizens or permanent residents of Australia (72.37%; n = 1011). Most respondents reported having been born in SEA (36.74%; n = 547), followed by NEA (29.35%; n = 437) and SSA (24.38%; n = 363) (9.54% of the sample did not specify a country/region of birth). Almost half (47.29%; n = 637) of respondents who reported a country of birth were born in a country with an HIV prevalence of less than 1%, 10.76% (n = 10.76) were born in a country with an HIV prevalence of 1.00–2.49%, and 9.36% (n = 80) were born in a country with an HIV prevalence over 2.5% (the remaining 32.59% were from countries for which HIV prevalence data are not available, namely, mainland China, Brunei, Japan, North Korea, South Korea, and Taiwan). The mean Socio-Economic Indexes for Areas (SEIFA) decile (lowest representing the most disadvantaged group), based on reported postcode of residence, was 5.69 (SD 2.68; 247 respondents did not provide postcode information to enable a SEIFA decile to be calculated). Detailed survey demographics are set out in Table 1. 

Twenty-three individuals participated in the qualitative component of this study, and their gender and region of birth are summarized in Table 2. 

### 3.2. Knowledge, Attitudes and Practices Regarding HIV Prevention

#### 3.2.1. Pre-Exposure Prophylaxis

Ninety-three percent (n = 1387) of survey respondents reported that they had heard of HIV and/or AIDS (n = 22 did not provide a valid response). Of respondents who had heard of HIV/AIDS and answered the question to ascertain knowledge of pre-exposure prophylaxis (PrEP) (n = 1366), 15.59% (n = 213) correctly indicated that there are “medicines that people can take BEFORE SEX to protect themselves against HIV”. As shown in Table 3, PrEP knowledge was significantly higher among: men who reported being sexually attracted to other men (MSM) compared to men who were not attracted to men (68.57% compared to 13.88%); respondents who identified as men (19.87%) compared to those who identified as women (12.97%); SSA-born respondents (22.05%) compared to those born in NEA (11.19%) and SEA (14.31%); and respondents aged 18–29 years (18.21%) and 30–39 years (20.90%) compared to respondents 40+ years. 

In response to a follow-up question asking survey respondents to name the medicine that can be taken before sex, less than one-fifth (17.37%; n = 37) provided a correct name (e.g., variation of PrEP, brand name or chemical name); with the remainder either indicating they did not know the name (52.11%; n = 111), skipped the question (2.35%; n = 5), said that they knew the name without providing the name (18.31%; n = 39), or provided an incorrect name (11.27%; n = 21). 

#### 3.2.2. Undetectable = Untransmittable (U = U)

Only 7.01% (n = 96) of survey respondents who had heard of HIV/AIDS and answered the question to ascertain knowledge of U = U (n = 1369), correctly indicated that it is “safe to have sex without a condom with someone who has VERY LOW amounts of HIV in their blood”; however, as noted in the paper in which survey pretesting results are presented, the low proportion of correct answers may reflect the fact that respondents may have found the wording of the question ambiguous [15]. 

#### 3.2.3. Sexual Behaviors and Condom Use

The majority (55.92%; n = 656) of survey respondents who provided information regarding number of sexual partners in the past 12 months reported having anal or vaginal intercourse with one partner, 31.80% (n = 373) reported no intercourse, 10.66% (n = 125) reported intercourse with at least two people, and 1.62% (n = 19) selected the option indicating they ‘could not recall’ the number of people with whom they had had intercourse in that timeframe. Among respondents who reported intercourse and who answered the question regarding the relationship status of their last sexual partner, 89.49% (n = 715) reported they most recently had sex with a person with whom they were in a ‘committed relationship’ (e.g., husband, wife, boyfriend, girlfriend). For MSM specifically, 9.75% (n = 4) reported no intercourse in the past 12 months, 34.15% (n = 14) reported intercourse with one partner, 56.10% (n = 23) reported intercourse with two or more partners, and 44.74% (n = 17) of those who were sexually active reported their last sexual encounter was with a ‘committed’ partner.

Condom use during their most recent sexual encounter was reported by 34.00% (n = 274) of the sexually active survey respondents who answered the question. Table 4 shows reported condom use was significantly higher among those whose most recent sexual encounter was with a non-committed partner (e.g., casual partner, sex worker), those who reported sex with two or more partners in the previous 12 months, those born in NEA and MSM. 

Reasons for not using condoms at last sexual encounter are summarized in Table 5, which summarizes reasons provided by people who reported sex with one committed sexual partner in the past 12 months with reasons provided by people who either had more than one sexual partner in the last 12 months or whose last sexual encounter was with a casual partner. For these groups, the most commonly cited reasons were: “My partner and I trust each other” (45.45%; n = 230); “My partner and I don’t have any illnesses that can be passed on through sex” (26.26%; n = 184); “My partner doesn’t like the way they feel” (12.06%; n = 61); “My partner did not want to use one” (14.82%; n = 75); “I don’t like the way they feel” (12.45%; n = 63); and “I did not want to use one” (17.00%; n = 86). A significantly higher proportion of people who either had more than one sexual partner in the last 12 months or whose last sexual encounter was with a casual partner or sex worker selected the latter two reasons, compared to people who only reported one committed sexual partner; the inverse was true in relation to the reason “My partner and I trust each other”. Trying to get pregnant was also a commonly cited reason (12.45%; n = 63) for respondents who reported one committed sexual partner. 

Reasons related to condom access or cultural factors were infrequently cited (Table 5): “My partner and/or I could not afford one” (0.99%; n = 5); “My partner and/or I did not know where to get one” (0.59%; n = 3); “It was against my or my partner’s culture or religion” (2.37%; n = 12). In open text ‘other’ reasons, nine respondents listed “PrEP” as their reason for not using a condom at last sexual encounter. 

One hundred and seven respondents (7.19% of the total sample) reported sex overseas (specific countries not specified) with at least one person who lives outside of Australia during travel since January 2018; of those who responded to the question on condom use during overseas sexual encounters, less than half (42.86%; n = 45) indicated that they always used condoms during these encounters. 

### 3.3. Knowledge, Attitudes and Practices Regarding HIV Testing 

#### 3.3.1. Timing of Last Test for STIs or BBVs

Of the respondents who answered the question about the recency of their last test for STIs and/or BBVs, less than one-third (31.33%; n = 449) reported being tested in the last two years, a similar proportion (31.96%; n = 458) reported never having been tested and 13.68% (n = 196) ‘could not recall’ the timing. Reasons for not testing for any STI or BBV in the last two years are set out in Table 6. The most reported reason was ‘I did not do anything to put me at risk’ (64.00%; n = 496); however, qualitative interviews revealed that risk perception can be influenced by the experience of migration, which may not be consistent with actual risk. Several interview participants described a perception that HIV was not a material risk in Australia due to low prevalence and perceptions of health system quality: 

*The first thing, I suppose we, the refugees, when we first came, we thought this place is safe, because some of these STDs are not existing here, because of the good health services that people are giving to people*.(SSA man, Western Australia, focus group)

*When we had the [pre-migration] interview, we got tested for HIV, syphilis, hepatitis A, B, and C, and gonorrhea. All of this, and chest x-ray to make sure you haven’t got TB and all that. So I even said to my husband, “Maybe where we’re going, nobody dies.”*.(SSA woman, Western Australia, focus group)

*I never really seriously thought about [blood-borne viruses], any risk in Australia … I think because it seems really controlled well … if the Australian government test every single migrant that means—like, I wouldn’t say no infections, but more, how do I say? Less chance*.(NEA man, Queensland, interview)

The next most reported reason for not testing was ‘I did not have any symptoms’ (30.45%; n = 236), although almost half of respondents (43.22%; n = 102) who chose this reason had indicated that they were aware that “a person can have an STI without any symptoms” in relation to an earlier knowledge question. Qualitative interviews indicated many participants did not access health services in the absence of serious symptoms in their countries of origin; for some, these practices continued post-migration (at least initially): 

*I think at the beginning our family and many of our parents’ friends, … they don’t go to doctors and hospital unless they’ve broken their arms or something like serious. I don’t think they used to go to doctors as much as they do now when they first arrived*.(NEA man, Queensland, interview)

*If they don’t feel anything wrong, they don’t go to the GP … They only go when they feel they’re sick*.(SEA woman, Queensland, interview)

*[In my birth country] you don’t really go to the doctor and get checked out unless it’s serious … Here … you don’t feel the need to go to a doctor unless they start to feel sick*.(SEA woman #1, Western Australia, interview)

*Only when we [are] sick, just go to see a doctor, and that’s it. We don’t do that kind of thing [preventative health services such as vaccinations] so I’m not really familiar when I came here*.(SEA woman, Western Australia, interview)

Only 1.68% (n = 13) respondents cited being “too embarrassed” as a reason for not seeking STI/BBV testing in the last two years; however, shame and stigma around STIs and BBVs was a recurring theme in the qualitative data. An SEA interviewee noted that:

*[People from my country are] … afraid of being judged. It’s all about your thoughts and you’re afraid people will judge you, but it might not be the case. It’s just you get in your own head and you would be afraid of being judged. They would search it up of course, they would Google because it’s anonymous. But to actually come up to a person … or to a doctor and ask them, “How do I get - I want to get myself checked, how do I do it?” I don’t think they would ever do it*.(SEA woman, Western Australia, interview)

Similarly, a SSA focus group participant spoke of *a “fear of being looked like you are not a very good person”* (SSA, Western Australia, focus group discussion), and a SEA woman noted that she would avoid seeking a test from her *“Burmese female doctor … I would go somewhere else … I don’t like people to judge me”* (SEA woman, Western Australia, interview). 

#### 3.3.2. Type of Last STI/BBV Test

Of the 407 survey respondents who reported having been tested for any STI/BBV in the last two years and who answered the question “What was your most recent STI and/or BBV test for? (Tick as many as apply)”, less than half (45.95%; n = 187) reported being tested for HIV, 14.25% (n = 58) responded “I don’t know - it was a blood test”, 15.72% (n = 64) responded “I don’t know – it was a blood and urine test”, and 6.14% (n = 25) responded “Other”. Open text descriptions accompanying the choice “Other” included: “Annual overall blood test”, “General check-up”, “General blood test” and “Health check”. 

The high proportion of respondents unaware of whether they were tested for HIV specifically is consistent with the finding that less than half (41.24%; n = 572) of respondents who had heard of HIV/AIDS were aware that an HIV test is not “done whenever someone has a blood test in Australia.” According to one interview participant, *“many people don’t even know that there’s different blood testing … they don’t know it is for HIV or it is for something else … [P]eople think, I’ve done blood tests, but they do not know what they were tested for”* (SEA man, Queensland, interview). A similar sentiment was expressed by a SSA-born man: *“I went to the doctor for a check-up, blood taken … and something that I didn’t understand was whether all these things have been tested or not … whether I need to tell the doctor specifically I need to be tested for AIDS”* (SSA man, Western Australia, interview). 

A potential source of confusion may relate to the way in which tests are conducted in migrants’ countries of origin. One interviewee recalled an experience of being tested for STIs and BBVs outside of Australia without having requested those tests or being informed that they were being conducted: 

*I didn’t even know I was getting checked until I got the results … I was like okay, just go from one room to another and get myself checked, it’s like do whatever you want. I didn’t even know until the results came and I got the results, I’m like “Huh, I got an STD check”*.(SEA woman, Western Australia, interview)

#### 3.3.3. Reason for Last HIV Test and Attitudes to Testing

Among respondents who indicated that they had been tested for HIV in the last two years and who provided a reason for testing (n = 164), the most cited reason for testing was “It was part of my regular health check” (35.37%; n = 58) (note that this finding ought to be read in light of the data in 3.2.1 which indicates confusion about the relationship between HIV testing and other blood testing). Other frequently cited reasons include: “I wanted to know if I had a sexually transmitted infection or a blood-borne virus” (20.12%; n = 33); “I like to get regular STI/BBV tests” (15.85%; n = 26); and “It was a requirement for my work/study” (15.24%; n = 25). 

Only a small proportion of respondents (7.93%; n = 13) indicated that they tested for HIV at the suggestion of their doctor/nurse; however, a majority of those who answered the question on attitudes to opportunistic testing (“How would you feel if a doctor in Australia offered you STI and BBV tests during an appointment without you requesting any of these tests?”) responded “Okay—STI and BBV testing is normal” (50.40%; n = 692). Approximately one-fifth reported that they would be “worried” (21.63%; n = 297), a similar proportion said they would be “surprised” (21.78%; n = 299) and smaller proportions reported that they would be “offended” (10.56%; n = 145), “embarrassed” (4.52%; n = 62) or “relieved” (6.92%; n = 95) (note that respondents could choose more than one option). 

Similarly, the most frequently cited reactions to the question “If a close friend in Australia told you that they were going to get tested for STIs and BBVs, how would you feel?” were: “Supportive—I am here if they need my help” (43.11%; n = 610); “Fine—it’s none of my business” (38.94%; n = 551); “Worried—I hope they are okay” (21.77%; n = 308); and “Proud—it’s the responsible thing to do” (17.53%; n = 248). Smaller proportions of respondents to the question indicated that they would be “shocked” (4.95%; n = 70) or “disappointed” (2.54%; n = 36). 

#### 3.3.4. Characteristics of Respondents Who Reported HIV Testing

Table 7 shows the results of bivariate analyses of predictors of self-reported HIV testing in the last two years. Statistically significant differences in testing were observed with respect to: region of birth (testing rates higher among SSA-born respondents); gender (testing rates higher among men); age (testing rates higher among those aged under 40 years), SEIFA decile (testing rates highest in decile 3) and sexuality (testing rates highest among MSM). Testing rates were also significantly higher among those who reported more sexual partners, those whose last sexual encounter was with a casual partner and those who correctly answered HIV knowledge questions.

## 4. Discussion

The MiBSS findings highlight potential points of intervention to address HIV disparities between migrants and domestic-born Australians. With respect to prevention, the introduction and uptake of PrEP has been described as a key factor in reducing HIV notifications in Australia [24]. However, the survey data revealed low levels of PrEP knowledge in this sample, which is as expected given that PrEP promotion in Australia has generally prioritized gay and other MSM. While knowledge was significantly higher among men who reported sexual attraction to other men, it is noteworthy that age-standardized rates of new HIV diagnoses attributed to heterosexual contact are higher among SEA- and NEA-born men, compared to Australian-born men [25]. Low PrEP awareness in samples of predominately non-MSM migrants has been reported in other settings [26,27,28,29]. A needs assessment conducted in the United States found that PrEP campaign assets tended to feature images of gay men and noted that the absence of images of African people meant that the “population did not feel included as potential beneficiaries of PrEP” [29] (p. 140). Other recommendations for improving PrEP awareness in migrant contexts include educational sessions involving community workers and clinicians, and peer-led educational initiatives [30,31]. 

Condom use at last sexual encounter was reported by 34.00% of sexually active survey respondents who answered the question; this is broadly consistent with the Second Australian Study of Health and Relationships (ASHR2) which found that 36.9% of men and 29.9% of women who spoke a language other than English at home used a condom during their most recent instance of vaginal intercourse, compared with 24.6% of men and 20.7% of women who only spoke English at home [32]. Among MiBSS respondents whose last sexual encounter was not with a committed partner, 56.63% reported using condoms during that encounter; this is similar to ASHR2’s general population survey which found that 64.8% of men and 55.8% of women used a condom during their most recent casual sexual encounters [32]. The main reasons cited by MiBSS respondents for not using condoms related to personal/partner preference, and ideas about risk and pleasure, rather than structural barriers such as cost and access. A study involving West African refugees living in Western Australia found that peer education can be effective at changing attitudes towards condom use [33]. 

The majority of the MiBSS sample who reported sex during travel outside of Australia reported only sometimes or never using a condom during those encounters. This is significant in that over a quarter of people born outside of Australia with newly acquired HIV attributable to heterosexual sex are likely to have acquired the virus overseas [2]. International travel to visit friends and relatives has been identified as a risk factor for the acquisition of sexually transmissible infections, including HIV [34,35]. Data collected by Mullens and colleagues during a community forum involving African Australians found that “there was strong support during the forum that information and advice needed to be provided to travelers on the risks of HIV transmission when returning to home countries” [30] (p. 7). While there is some evidence to suggest that providing clients with literature on sexually transmissible infections during pre-travel health consultations is associated with more consistent condom use during travel, more research on intervention effectiveness is needed [36].

With respect to testing, the MiBSS study helps to overcome the limitations associated with the fact that information on country of birth is not routinely collected or reported in Australian laboratory testing data (unless a positive HIV diagnosis is returned). The self-reported testing data suggests that guidance in the National HIV Testing Policy is not being followed [37]. Under the Policy, HIV testing is indicated for “[p]eople who have recently changed partners, … or who have had multiple partners since their last HIV test” [37] (p. 12); however, results of the survey reveal that less than half of respondents with casual or multiple sexual partners had been tested for HIV in the last two years. It is possible that the proportion tested is even lower than reported given the commonly held belief (noted in both survey and interview data) that HIV testing is routinely included in any blood test. Consequently, some individuals may be declining or not requesting HIV tests (and, in the context of this survey, may have reported that they were recently tested for HIV) based on the erroneous assumption that testing is conducted in the course of other bloodwork. 

Calls have been made for the introduction of opt-out HIV testing in higher-prevalence settings as a means of increasing testing rates [38]. Alternative suggestions to increase HIV testing uptake in migrant contexts include more direct offers for opt-in testing [39]. It is noted that Australian testing guidelines specifically recommend opportunistic HIV testing for people from high-prevalence countries; however, this may not be occurring in practice [36]. In the MiBSS sample, few respondents reported having undertaken HIV testing because a doctor or nurse had offered it to them. The literature contains mixed findings on the acceptability of provider-initiated testing in migrant contexts. Some qualitative studies suggest that migrants may be offended by offers of HIV testing because it implies that everyone from a migrant background is at risk of HIV [40,41,42,43]. However, other studies suggest that provider-initiated testing can be acceptable [44,45,46,47]. In a Belgian study, Manirankunda and colleagues noted that SSA participants “perceived doctors as a health authority knowing what is best for the patient” [45] (p. 589), and a study of Latin American migrants living in Spain found that most participants preferred being asked (over having to ask) for a test [46]. In the MiBSS sample, most respondents indicated that they would be ‘okay’ with a provider-initiated offer of testing, while only 10.56% reported that they would be offended. 

Consideration should be given to developing, promoting and evaluating resources and workforce capacity building initiatives to ensure that health providers feel confident in offering testing in sensitive ways that do not reinforce feelings of stigma or discrimination [48]. For example, the Your Cultural Lens training resource developed by the Western Australian Department of Health aims to support health practitioners to engage in cross-cultural communication about sexually transmissible infections and blood-borne viruses [49]. Additionally, Loos and colleagues report high acceptability of a project in which general practitioners were provided with counseling guidelines around provider-initiated HIV testing for SSA-born clients, including information on cultural context and tips for communication [50]. 

The results suggest that interventions designed to upskill health professionals must be accompanied by initiatives to educate people from migrant backgrounds to engage with health care on a routine basis. Almost half of the respondents who indicated that they did not get an STI/BBV test in the last two years because they did not have any symptoms were nevertheless aware that STIs can be asymptomatic. Consistent with other studies, the qualitative findings suggest that perceptions of risk and unfamiliarity with preventative health help-seeking may explain the apparent anomaly between HIV knowledge and HIV testing practices [30,45,51,52,53,54]. The literature suggests that there is a need for investment in community-based programs to improve literacy about the availability and nature of preventative health services among people from migrant backgrounds; this includes designing and delivering messages in culturally safe ways that go beyond simple in-language translation/interpretation to account for the universally stigmatized nature of STIs/BBVs [55]. An alternative to encouraging migrants to attend health clinics may be to widen testing options, including access to at-home testing kits and outreach services [56,57]. Research suggests that programs which utilize bilingual cultural workers and peer health educators can be effective for increasing STI/BBV knowledge and help-seeking behaviors in migrant contexts [58]. Health promotion initiatives at educational institutions such as universities have also been proposed as a potential means of improving STI/BBV health help-seeking among international students [59]. 

Limitations of this study relate to the non-probability sampling approach. While the approach is common in cross-sectional sexual health surveys of migrant populations (as described in the Methods above), it does introduce a risk of selection bias with some studies showing that ‘volunteers’ are more likely to have less conservative sexual attitudes and behaviors than those recruited randomly [60]. Similarly, the fact that respondents were able to skip questions means that those who provided answers to sensitive questions may differ from those who did not [60]. However, it is noteworthy that our findings are broadly consistent with other qualitative and quantitative studies of migrant populations in Australia and internationally, as set out in the preceding Discussion. 

## 5. Conclusions

This study represents the first attempt to assess HIV-related knowledge, risk behaviors and testing practices in a national sample of migrants living in Australia; as such, it represents an important contribution to the evidence base, given the national HIV Strategy’s focus on improving HIV outcomes in culturally and linguistically diverse populations. The study also offers an example for other countries seeking to design and implement national sexual health and BBV surveys in migrant populations. Currently, there are few examples of national migrant sexual health and BBV surveys notwithstanding the Sustainable Development Goals’ target of universal access to sexual and reproductive health services, [61] and the UCL Commission on Migration and Health’s recognition of sexual and reproductive health as an area requiring attention [62].

The results point to several areas that require intervention to mitigate against widening disparities in HIV outcomes between migrant and non-migrant Australians. In particular, effort is needed to improve knowledge of PrEP, redress misconceptions around testing and encourage safer sex practices during casual sexual encounters and in the course of international travel. Workforce capacity building is also needed to increase opportunistic testing and adherence to national testing guidelines. 

These data provide an evidence base to inform policy- and service-level improvements, and a baseline from which to assess whether strategic interventions are proving effective at changing knowledge and practices in migrant populations. Periodic administration of the survey is recommended to enable trends to be observed over time, and to assist in the identification of new strategic priorities. 

## Figures and Tables

**Table 1 ijerph-20-04347-t001:** Number of survey respondents, by region of birth, age, gender and time in Australia.

Place of Birth	Age (Years)						Gender			Length of Time in Australia (Years)					SEIFA Decile					
	18–29	30–39	40–49	50–59	60+	N/A	Man	Woman	N/A and Other	≤9	10–19	20–29	30+	N/A	1–2	3–4	5–6	7–8	9–10	N/A
SSA (n = 363)																				
Eastern ^1^ (n = 270)	86	91	60	25	8	0	124	137	9	85	161	9	3	12	55	29	70	47	14	55
Western ^2^ (n = 53)	18	11	9	10	3	2	24	38	1	21	21	4	1	6	21	4	12	3	2	11
Central ^3^ (n = 24)	10	7	5	1	0	1	15	9	0	11	13	0	0	0	8	4	3	1	0	8
Southern ^4^ (n = 16)	1	7	7	1	0	0	4	11	1	6	6	1	2	1	2	1	5	2	5	1
SEA (n = 547)																				
Cambodia (n = 27)	12	6	7	1	1	0	16	10	1	18	6	0	3	0	7	5	3	2	4	6
Indonesia (n = 165)	31	37	46	34	15	2	50	115	0	70	50	21	18	6	15	24	33	40	37	16
Malaysia (n = 66)	21	14	9	7	15	0	24	40	2	26	19	4	14	3	3	10	14	16	17	6
Myanmar (n = 27)	8	5	4	4	6	0	11	15	1	14	4	3	5	1	9	2	6	2	5	3
Philippines(n = 104)	22	25	17	18	20	2	30	71	3	46	18	7	28	5	19	20	23	27	11	4
Singapore (n = 38)	12	3	5	7	10	1	10	28	0	14	9	6	8	1	1	5	9	12	11	0
Thailand (n = 22)	4	10	8	0	0	0	9	12	1	8	11	1	2	0	4	3	2	6	5	2
Vietnam (n = 95)	28	17	19	12	15	4	44	48	3	38	18	11	26	2	32	3	16	10	11	23
Other (small cells combined) (n = 3)	1	2	0	0	0	0	2	1	0	0	1	2	0	0	1	0	0	0	2	0
NEA (n = 437)																				
Mainland China (n = 179)	73	51	29	16	8	2	74	104	1	123	33	11	6	6	9	35	55	42	22	16
Hong Kong / Macau (n = 14)	4	6	1	2	1	0	3	10	1	9	5	0	0	0	0	0	5	4	4	1
Taiwan (n = 41)	5	20	8	2	6	0	11	30	0	21	9	3	6	2	3	5	14	12	3	4
Japan (n = 23)	4	5	13	1	0	0	3	20	0	8	13	1	0	1	3	3	5	5	5	2
Korean peninsula (n = 180)	35	80	55	7	3	0	33	147	0	78	80	19	2	1	11	17	37	54	55	6
Unknown (n = 142)	21	32	11	19	5	54	37	44	61	23	39	2	9	69	16	9	17	10	7	82
TOTAL (n = 1489)	396	429	313	167	116	68	524	880	85	619	516	104	134	116	219	179	329	295	220	247

^1^ Eastern sub-Saharan Africa is defined here as Tanzania, Kenya, Uganda, Rwanda, Burundi, Sudan, South Sudan, Eritrea, Ethiopia, Somalia, Comoros, Mauritius, Seychelles, Mozambique, Madagascar, Malawi, Zambia, Zimbabwe. ^2^ Western Sub-Saharan Africa is defined here as Benin, Burkina Faso, Cape Verde, The Gambia, Ghana, Guinea, Guinea-Bissau, Ivory Coast, Liberia, Mali, Mauritania, Niger, Nigeria, Senegal, Sierra Leone, Togo. ^3^ Central sub-Saharan Africa is defined here as Angola, Cameroon, Central African Republic, Chad, Democratic Republic of Congo, Republic of Congo, Equatorial Guinea, Gabon, Sao Tome and Principe. ^4^ Southern sub-Saharan Africa is defined here as Botswana, Eswatini, Lesotho, Namibia, South Africa.

**Table 2 ijerph-20-04347-t002:** Characteristics of participants in qualitative component of study.

Participant ID ^1^	Place of Birth	State of Residence	Gender	Length of Time in Australia (Years)
Participant 1 ^1^	SSA	Western Australia	Woman	11
Participant 2 ^1^	SSA	Western Australia	Woman	16
Participant 3 ^1^	SSA	Western Australia	Woman	18
Participant 4 ^1^	SSA	Western Australia	Woman	Not stated
Participant 5 ^1^	SSA	Western Australia	Man	Not stated
Participant 6	SEA	Western Australia	Woman	7
Participant 7	SEA	Western Australia	Woman	3
Participant 8	NEA	Western Australia	Woman	11
Participant 9	SSA	Western Australia	Woman	5
Participant 10	SSA	Western Australia	Man	17
Participant 11	SEA	Western Australia	Man	16
Participant 12	SEA	Queensland	Man	6
Participant 13	NEA	Queensland	Woman	14
Participant 14	NEA	Queensland	Man	18
Participant 15	NEA	Queensland	Man	25
Participant 16	SEA	Queensland	Woman	10
Participant 17	SEA	Queensland	Man	22
Participant 18	NEA	Queensland	Man	4
Participant 19	SSA	Queensland	Man	8
Participant 20	SEA	Queensland	Man	10
Participant 21	NEA	Queensland	Man	20
Participant 22	NEA	Queensland	Man	3
Participant 23	NEA	Queensland	Man	4

^1^ Indicates focus group discussion participants (all others participants were interviewed).

**Table 3 ijerph-20-04347-t003:** Responses to PrEP knowledge question (Are there any medicines that people can take BEFORE SEX to protect themselves against HIV?) by demographic variables.

	Responses, n(%)		
Demographic Variables	Yes (Correct)	No/I Don’t Know	*p*-Value
MSM (n = 46)	32 (69.57)	14 (30.43)	<0.001
Non-MSM men (n = 418)	58 (13.88)	360 (86.12)	
Men (n = 478)	95 (19.87)	383 (80.13)	0.001
Women (n = 817)	106 (12.97)	711 (87.03)	
NEA (n = 411)	46 (11.19)	365 (88.81)	<0.001
SEA (n = 503)	72 (14.31)	431 (85.69)	
SSA (n = 331)	73 (22.05)	258 (77.95)	
18–29 year olds (n = 368)	67 (18.21)	301 (81.79)	<0.001
30–39 year olds (n = 402)	84 (20.90)	318 (79.10)	
40–49 year olds (n = 291)	38 (13.06)	253 (86.94)	
50–59 year olds (n = 152)	11 (7.24)	141 (92.76)	
60+ year olds (n = 99)	6 (6.06)	93 (93.94)	

**Table 4 ijerph-20-04347-t004:** Responses to condom use question (Did you use a condom the MOST RECENT time you had sex?), by sexual practices, birth region and sexual preference.

	Responses, n(%)		
Sexual Practice	Yes	No/Can’t Recall	*p*-Value
Most recently had sex with committed partner (n = 704)	222 (31.53)	482 (68.47)	<0.001
Most recently had sex with a non-committed partner (n = 83)	47 (56.63)	36 (43.37)	
Only one sex partner in past 12 months (n = 644)	192 (29.81)	452 (70.19)	<0.001
Multiple sex partners in past 12 months (n = 139) ^1^	72 (51.80)	67 (48.20)	
NEA (n = 296)	131 (44.26)	165 (55.74)	<0.001
SEA (n = 272)	88 (32.35)	184 (67.65)	
SSA (n = 180)	40 (22.22)	140 (77.78)	
MSM (n = 38)	19 (50.00)	19 (50.00)	0.049
Non-MSM men (n = 166)	84 (33.60)	166 (66.40)	

^1^ Includes those who could not recall number of sexual partners.

**Table 5 ijerph-20-04347-t005:** Responses to reason for condom non-use (Why did you NOT use a condom the MOST RECENT time you had sex?), by number and type of partners (n = 506).

	Respondents n (% of Sub-Group)		
Reasons ^1^	One Partner in Past 12 Months (Committed) (n = 424)	Multiple Partners ^2^ in Past 12 Months or in Non-Committed Relationship with Most Recent Partner (n = 82)	*p*-Value
My partner and/or I did not have one	25 (5.90)	10 (12.20)	0.040
My partner and/or I could not afford one	3 (0.71)	2 (2.44)	0.147
My partner did not want to use one	62 (14.62)	13 (15.85)	0.774
I did not want to use one	62 (14.62)	24 (29.27)	0.001
My partner and/or I did not know where to get one	1 (0.24)	2 (2.44)	0.017
My partner doesn’t like the way they feel	47 (11.08)	14 (17.07)	0.127
I don’t like the way they feel	46 (10.85)	17 (20.73)	0.013
My partner or I was trying to get pregnant	56 (13.21)	7 (8.54)	0.241
It is against my or my partner’s culture or religion	10 (2.36)	2 (2.44)	0.965
My partner and I don’t have any illnesses that can be passed on through sex	161 (37.67)	23 (28.05)	0.087
My partner and I trust each other	206 (48.58)	24 (29.27)	0.001
Another reason	37 (8.73)	9 (10.98)	0.517

^1^ Respondents could choose as many reasons as apply. ^2^ Includes those who could not recall the number of partners.

**Table 6 ijerph-20-04347-t006:** Responses to reason for no STI/BBV test in last two years (Why did you NOT have an STI or BBV test in the last two years?) (n = 775).

Category	Reasons ^2^	Survey Respondents n(%) ^1^
Risk assessment	I did not do anything to put me at risk	496 (64.00)
Symptoms	I did not have any symptoms	236 (30.45)
Priorities	I did not think it was important	70 (9.03)
	I did not have the time to get tested	38 (4.90)
Service access barriers	I didn’t know where to get one	50 (6.45)
	I could not afford extra tests	22 (2.48)
	I don’t like needles/blood tests	22 (2.48)
	I could not get to a service/clinic	11 (1.42)
Psychological	I was too embarrassed	13 (1.68)
	I was scared about the result	12 (1.55)
Other	Another reason	64 (8.26)

^1^ Respondents who indicated they were tested ‘never’ or ‘more than two years ago’ and who provided a reason. ^2^ Respondents could choose as many reasons as apply.

**Table 7 ijerph-20-04347-t007:** HIV testing, by socio-demographic factors, HIV knowledge, and sexual practices.

Variable	Tested for HIV in Last Two Years		
	n(%)		*p*-Value
Region of Birth (n = 1125)			
NEA	54 (13.88)		<0.001
SEA	53 (11.65)		
SSA	71 (25.27)		
Gender (n = 1163)			
Man	79 (18.37)		0.050
Woman	103 (14.05)		
Age in years (n = 1176)			
18–29	51 (15.69)		<0.001
30–39	91 (24.66)		
40–49	37 (13.55)		
50–59	4 (3.33)		
60+	1 (13.90)		
Years in Australia (n = 1141)			
0–9	94 (17.94)		0.054
10–19	64 (15.02)		
20–29	14 (16.67)		
30+	8 (7.48)		
Visa status (n = 1155)			
Permanent resident/citizen	129 (15.83)		0.919
Other visa	53 (15.59)		
SEIFA decile (n = 1041)			
1	21 (21.21)		0.049
2	10 (14.49)		
3	6 (26.09)		
4	10 (8.00)		
5	24 (13.41)		
6	24 (25.53)		
7	16 (16.49)		
8	26 (15.85)		
9	21 (17.07)		
10	12 (17.65)		
Languages spoken at home (n = 400)			
Language(s) other than English only	42 (21.43)		0.265
Only English	24 (28.57)		
English and at least one other language	16 (18.60)		
MSM status (n = 1151)			
MSM	25 (58.14)		<0.001
Not MSM	155 (13.99)		
Aware of PrEP (n = 1122)			
Yes	60 (33.15)		<0.001
No	119 (12.65)		
Aware STIs can be asymptomatic (n = 1063)			
Yes	155 (23.07)		<0.001
No	26 (6.65)		
Aware HIV test not part of all blood tests (n = 1123)			
Yes	86 (18.07)		0.110
No	94 (14.53)		
Number of sexual partners in last year (n = 978)			
0	30 (9.68)		<0.001
1	85 (15.21)		
≥2	46 (42.20)		
Identity of last sexual partner (n = 758)			
Committed partner	109 (16.01)		<0.001
Non-committed partner	33 (42.86)		
Condom used at last sexual encounter (n = 686)			
Yes	54 (22.98)		0.117
No/can’t recall	81 (17.96)		
Travelled overseas since January 2018 (n = 1183)			
Yes	129 (16.88)		0.138
No	57 (13.60)		

## Data Availability

Data supporting reported results can be obtained by contacting the corresponding author.

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
