# Peer review of "HIV-Related Knowledge and Practices among Asian and African Migrants Living in Australia: Results from a Cross-Sectional Survey and Qualitative Study"

_ijerph, 2023, doi:10.3390/ijerph20054347_

Round 1
Reviewer 1 Report
In this paper the authors have performed a cross-sectional study and qualitative study of HIV knowledge among Asian and African migrants.
The study is well-performed. The manuscript is well-written and easy to follow. The study highlights knowledge gaps in the studied population. My only comment is that I feel the study is missing out on questions regarding "U=U ie undetectable equals untransmittable". It is too late to include this in the study but at least it could be commented on in then Discussion.
Author Response
Thank you for your kind and helpful comments. Please see attachment.

Reviewer 2 Report
The paper entitled “HIV-related knowledge and practices among Asian and African migrants living in Australia: results from a cross-sectional survey and qualitative study”, aims to answer the following questions: (1) What are Northeast Asia, Southeast Asia and Sub-Saharan Africa migrants’ knowledge, attitudes, and practices in relation to HIV prevention and HIV testing in Australia; and (2) What are the barriers to and enablers of HIV prevention and HIV testing practices among Northeast Asia, Southeast Asia and Sub-Saharan Africa migrants living in Australia?
At the methodological level, in 2019 preliminary qualitative research was conducted to understand how migrants in Australia perceive and engage with services and resources relating to sexually transmissible infections and blood-borne viruses. Accordingly, eighteen in-depth interviews and one focus group were audio-recorded and analyzed following a grounded theory approach. After considering the themes identified in the qualitative data an English-language survey instrument was developed. A survey sample comprised of 372 adults born in Northeast Asia, 372 adults born in Southeast Asia, and 372 adults born in Sub-Saharan Africa and living in Australia was formed.
The authors recruited 372 migrants from Northeast Asia, Southeast Asia and Sub-Saharan Africa respectively in order to enable inter-group variations to be explored at a 5% sampling error and a 90% confidence interval. However, this information is misleading. Inferential statistics able to draw inferences about the population data from sample data should derive from a sample representative of the entire population of Northeast Asia, Southeast Asia and Sub-Saharan Africa migrants living in Australia. However, a non-probability sampling technique was adopted. Therefore, sampling errors and confidence intervals cannot be calculated. Interviewing 372, 500 or 1000 migrants from Northeast Asia, Southeast Asia and Sub-Saharan Africa respectively does not allow the authors to quantify sampling errors and confidence intervals. Non-probability sampling techniques do not allow to draw inferences about the population of Northeast Asia, Southeast Asia and Sub-Saharan African migrants living in Australia.
Therefore, the authors should remove this sentence: “To enable inter-group variations to be explored at a 5% significance level and 90% power, the goal was to recruit a minimum sample of 1,116 (372 respondents per birth region). This sentence can be changed into this: “To explore inter-group variations 372 respondents per birth region were recruited”. The first sentence is misleading. The second sentence is accurate.
In this kind of studies, probabilistic samples representative of the entire population studied are not only impossible to obtain; but are also quite useless because of poor responses rate, etc. The authors reasoning about the limitations versus the strength of non-probability sampling techniques to study HIV-related knowledge and practices among Asian and African migrants living in Australia convinces me.
I find this paper interesting. This is an exploratory study of an important and non-researched issue in Australia: HIV-related knowledge, risk behaviours and testing practices in a national sample of migrants living in Australia. However, I found the conclusions extremely concise. I would recommend the authors write a more lengthy and discursive conclusion.
Author Response

(The authors gave the same response as above.)
